# Gate-tunable anomalous Hall effect in Bernal tetralayer graphene

Hao Chen[1,3], Arpit Arora[2,3], Justin C. W. Song [2,4] & Kian Ping Loh [1,4]

Large spin-orbit coupling is often thought to be critical in realizing magnetic order-locked charge transport such as the anomalous Hall effect (AHE). Recently, artificial stacks of two-dimensional materials, e.g., magic-angle twisted bilayer graphene on hexagonal boron-nitride heterostructures and dual-gated rhombohedral trilayer graphene, have become platforms for realizing AHE without spin-orbit coupling. However, these stacking arrangements are not energetically favorable, impeding experiments and further device engineering. Here we report an anomalous Hall effect in Bernal-stacked tetralayer graphene devices (BTG), the most stable configuration of four-layer graphene. BTG AHE is switched on by a displacement field and is most pronounced at low carrier densities. The onset of AHE occurs in tandem with a full metal to a broken isospin transition indicating an orbital origin of the itinerant ferromagnetism. At lowest densities, BTG exhibits an unconventional hysteresis with step-like anomalous Hall plateaus. Persisting to several tens of kelvin, AHE in BTG demonstrates the ubiquity and robustness of magnetic order in readily available and stable multilayer Bernal graphene stacks—a new venue for intrinsic non-reciprocal responses.

Electrical access to the valley degree of freedom in two-dimensional Dirac materials requires the simultaneous breaking of inversion and time-reversal symmetries. For instance, by irradiating $MoS_2$[1] or (inversion broken) gapped bilayer graphene with circularly polarized light[2], valleys can be directly addressed and detected via a valley selective Hall effect. Recently, intrinsic valley access in the absence of external magnetic field or circularly polarized light irradiation was demonstrated via orbital ferromagnetic ordering in moiré materials[3,4], and rhombohedral trilayer graphene[5,6], giving rise to an anomalous Hall effect (AHE). In these, valley polarization can be directly identified by the sign of the anomalous Hall response at zero magnetic field and switched hysterically[3–7]. However, consistently achieving these stacking configurations for device characterization is challenging[8]. In twisted bilayer graphene, AHE has only been found for twist angles that are simultaneously close to magic angle as well as aligned to a hexagonal boron nitride substrate[3,4,8], while rhombohedral stacking arrangements are not the thermodynamic ground state in naturally occurring graphene trilayers[9–11].

Here we uncover an AHE in Bernal-stacked tetralayer graphene (BTG), the most energetically stable allotrope of four-layer graphene[9,10], as illustrated in Fig. 1a. BTG has been found to possess unconventional electronic behavior, including gate-tunable Lifshitz transitions at moderate density[12], as well as topological phase transitions and helical states at high magnetic field[13]. In contrast to these previous works, we focus on the low-density and moderate displacement field region in BTG that displays pronounced valley-locked orbital magnetic moments and prominent density of states close to van Hove singularities (Fig. 1b, plotted using an eight-band model[14], see also Supplementary Information). These features interact to produce an AHE.

## Results

We fabricated dual-gated BTG in a Hall bar structure (see Fig. 1a for device schematic; see Methods and Supplementary Fig. 1 for device fabrication details); here graphite top and bottom gate electrodes were employed and a four-probe setup was used to measure the

[1]Department of Chemistry, National University of Singapore, Singapore, Singapore. [2]Division of Physics and Applied Physics, School of Physical and Mathematical Sciences, Nanyang Technological University, Singapore, Singapore. [3]These authors contributed equally: Hao Chen, Arpit Arora. [4]These authors jointly supervised this work: Justin C.W. Song, Kian Ping Loh. e-mail: justinsong@ntu.edu.sg; chmlohkp@nus.edu.sg

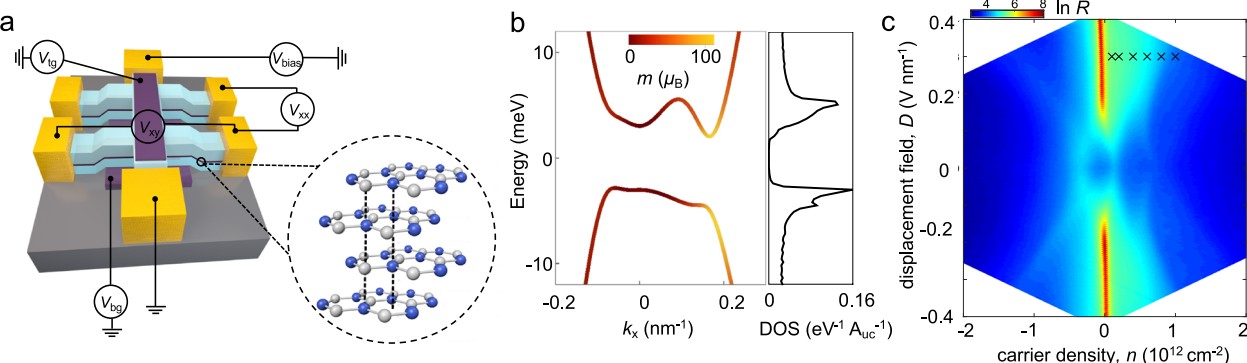

**Fig. 1 | Electronic properties of BTG. a** Schematic of a BTG Hall bar hetero-structure and measurement setup. Inset shows the BTG crystal structure where white and blue balls indicate A and B sites of the carbon atoms. **b** Energy spectrum and corresponding density of states (DOS) of BTG at finite interlayer potential difference $\Delta = 30$ meV. Here, $A_{uc}$ is the area of the unit cell. Color bar indicates the orbital magnetic moment which is concentrated around the band bottom; peaks in density of states correspond to van Hove singularities (vHs). The band structure was plotted using an 8-band model for BTG with unequal layer potential drop; an equal potential drop 8-band model for BTG, as well as a discussion of the magnetic moment is also described in the Supplementary Information. **c** Four-terminal resistance $R_{xx}$ as function of carrier density and displacement field. The cross symbols ('X') correspond to the $(n, D)$ conditions in Fig. 3e.

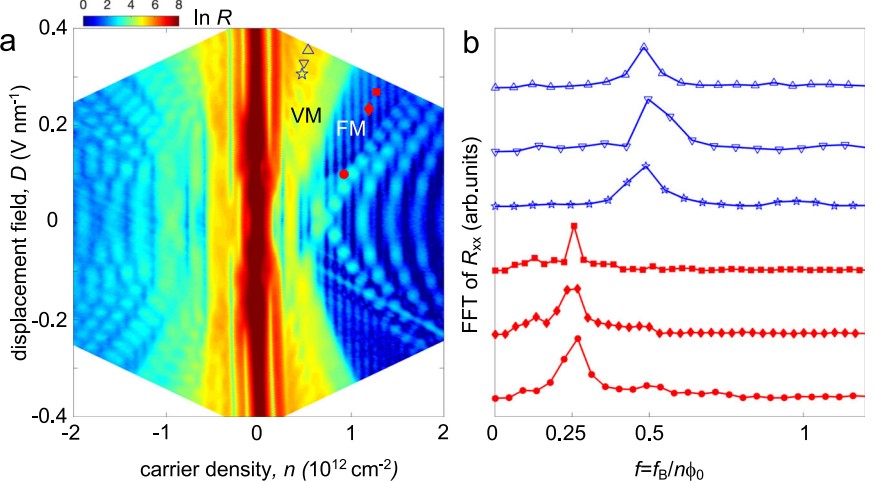

**Fig. 2 | Broken symmetries in BTG. a** Four-terminal resistance $R_{xx}$ as a function of displacement field and carrier density at $B = 1$ T. Selected points in VM and FM regions are marked with blue unfilled polygons (upward triangle, downward triangle and star) and red filled polygons (square, diamond, and circle), respectively. See also Supplementary Fig. 2 where the same resistance plot is shown with filling fraction indicated. **b** Fast Fourier transform (FFT) of $R_{xx}(1/B_\perp)$ measured at the $(n, D)$ points indicated by the corresponding symbols in three regions in **a**. Data are plotted in $f = f_B/(n\phi_0)$, where $f_B$ is the oscillation frequency (measured in tesla) and $\phi_0 = h/e$ is the magnetic flux quantum.

longitudinal resistance $R_{xx}$ and Hall resistance $R_{xy}$. The carrier density $n = [C_t(V_t\text{-}V_{t0}) + C_b(V_b\text{-}V_{b0})]/e$ and perpendicular displacement field $D = [C_t(V_t\text{-}V_{t0}) - C_b(V_b\text{-}V_{b0})]/2\epsilon_0$ can be independently tuned by applying voltages on top and bottom gates (see Fig. 1a). Here $C_t$, $C_b$, $V_t$, $V_b$, $e$ and $\epsilon_0$ denotes top and bottom gate capacitance and voltages, elementary charge and vacuum permittivity, respectively. Electrical measurements were carried out in a cryostat at 300 mK unless otherwise stated.

Figure 1c shows the variation of $R_{xx}$ with carrier density and displacement field at zero magnetic field, which comports with the unique characteristics of BTG transport[12]. For instance, at $D = 0$, $R_{xx}$ displays a resistance minimum at the charge neutrality point (CNP) and a multi-peak like structure as a function of carrier density; these peaks can be controlled by $D$ and have been attributed to gate-tunable Lifshitz transitions in BTG[12]. Similarly, $R_{xx}$ at the CNP first increases with $D$ and then saturates at larger $D$ ($> 0.2$ V/nm). These behaviors are signatures of Bernal-stacking order of BTG[12,15].

To investigate the fermiology of BTG, we measure $R_{xx}(n, D)$ at a small out-of-plane magnetic field of $B = 1$ T in Fig. 2a. In the following,

we focus on electron doping where oscillations in the resistance are most visible. Several striking features of $R_{xx}(n, D)$ are immediately apparent. First, and perhaps the most prominent, are vertical stripes that occur at uniform spacing of filling factor $\delta\nu = n\phi_0/B = 4$, where $\phi_0$ is the magnetic flux quantum. The degeneracy of 4 isospin flavors (2 valleys and 2 spins) is often associated with a "full" metal (FM) phase, see also Supplementary Fig. 2 for $R_{xx}(n, D)$ density plot with filling fraction labels. Additionally, superposed on top of the vertical stripes are diagonal fringes at moderate to high density that originate from the intersection between collections of Landau levels.

In contrast, at low density and moderate $D$ (see region labeled "VM" for valley metal and discussion below), an unusual stitch-like ripple pattern emerges. These ripples arise from a complex pattern of Landau levels and their intersections that disperse with D[12]. Strikingly, this region is marked conspicuously by the absence of the clear vertical lines with $\delta\nu = 4$ degeneracy spacing. To further investigate the features in this region, the period of quantum oscillations is examined in Fig. 2b as a function of $B$ that can reveal the $k$-space area of the enclosed orbits. Extracting the oscillation frequency, $f_B$, in the

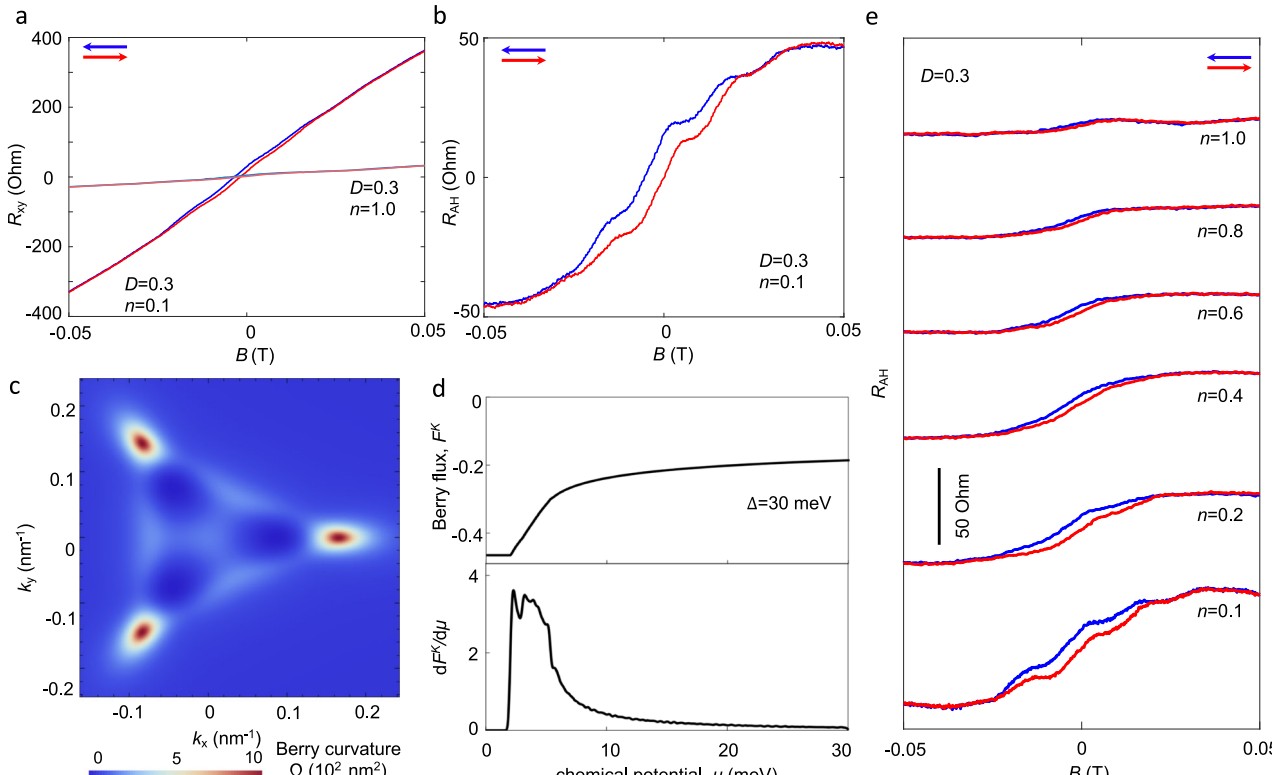

**Fig. 3 | Anomalous Hall effect in BTG. a** Hall resistance ($R_{xy}$) as a function of magnetic field at specific ($n$, $D$) conditions. **b** Anomalous Hall resistance $R_{AH}$ extracted from background under specified condition. **c** Berry curvature distribution in momentum space of BTG under perpendicular displacement field ($\Delta$=30 meV). **d** Berry flux (upper panel) and derivative of Berry flux (lower panel) as a function of chemical potential at finite interlayer potential difference $\Delta$=30 meV. In both panel c and d, Berry curvature was calculated numerically and plotted using the same 8 band model for BTG as Fig. 1; a similar peaked form of the Berry curvature can also be obtained using an equal potential drop 8-band model for BTG, see Supplementary Information. **e** Anomalous Hall resistance as a function of magnetic field at different carrier densities while keeping displacement field constant ($D$ = 0.3). Scale bar denotes 50 Ohm. $n$ and $D$ are in units of $10^{12}$ cm$^{-2}$ and V nm$^{-1}$, respectively. The ($n$, $D$) conditions are also denoted by cross symbols ('X') in the top part in Fig. 1c.

standard method[6,16] via Fourier analysis of $R_{xx}$ (1/$B$) (see Supplementary Fig. 3 for details) enables to quantify the $k$-space size of each orbit swept vs the total Fermi surface size (including all degeneracies) as $f = f_B / n\phi_0$; 1/$f$ indicates the degeneracy of the Fermi surface.

For clarity, we examine the quantum oscillations for several illustrative points in the "VM" region ("FM" region) denoted by the empty (solid) polygons respectively. In the "FM" region, we find $f \approx 1/4$ in agreement with the 4 fold-degenerate vertical stripes discussed above. In contrast, the "VM" region possesses $f \approx 1/2$ indicating a lowering of the isospin flavor degeneracy with an orbit area that is approximately half the total Fermi surface size. As a result, this behavior is reminiscent of a gate-tunable transition from a full metal to a half-metallic-like regime, similar to other isospin-polarized phase transitions reported in rhombohedral trilayer graphene[6] and bilayer graphene recently[16,17]. While the combination of an applied magnetic field (broken time-reversal) and $D$ field (broken inversion) are sufficient to lead to externally broken valley symmetry[18] and hence half-metallicity at low densities, as we now discuss, time-reversal remains spontaneously broken at $B = 0$. We denote this spontaneous time-reversal broken phase a valley metal state.

Importantly, spontaneous broken time-reversal symmetry in the "VM" region manifests as an AHE. Figure 3a displays the Hall resistance $R_{xy}$ as a function of applied $B$ field at $D = 0.3$ V nm$^{-1}$ at two different densities $0.1 \times 10^{12}$ cm$^{-2}$ (deep in the "VM" region) and $1.0 \times 10^{12}$ cm$^{-2}$ (outside the "VM" region). $R_{xy}$ for $1.0 \times 10^{12}$ cm$^{-2}$ exhibits a conventional linear in $B$ dependence with near identical forward (red) and backward (blue) $B$-field sweeps. In contrast, $R_{xy}$ for densities of $0.1 \times 10^{12}$ cm$^{-2}$ is hysteretic with a non-zero Hall resistance at $B = 0$ of several tens of

Ohms (see also Fig. 3b). Similar hysteretic behavior was also observed throughout the VM region, as well as for negative $D$; no hysteretic AHE was observed when $D = 0$ (see Supplementary Figs. 4 and 5).

BTG AHE can be better visualized by delineating the total Hall resistance into a conventional component that scales as applied $B$ and an AHE contribution[19]: $R_{xy}$ ($B$) = $\eta B + R_{AH}$ ($M$), where $\eta = 1/ne$ is the Hall coefficient and $M$ is the magnetization. Plotting $R_{AH}$ as a function of applied $B$ in Fig. 3b reveals a soft ramp-like profile: at low magnetic fields (up to 20–30 of mT), $R_{AH}$ increases rapidly before saturating at approximately 40–50 mT. $R_{AH}$ also highlights the hysteretic features of BTG AHE. In the small field window where hysteresis is most pronounced, the blue and red curves mirror each other to display the same stepped profile but displaced in the y-axis. This shift between forward and backward sweeps indicates that red (-) and blue (+) $R_{AH}$ curves track different ferromagnetic states (labeled, ± with magnetization $M_{\pm}$) as they evolve with applied $B$.

AHE in BTG can be naturally understood by examining the Berry curvature distribution in the bands of gated BTG. In valley K, the Berry curvature distribution (see Fig. 3c, also see Supplementary Figs. 10 and 12 for details) is peaked around the band bottom; valley K' possesses the same distribution but with an opposite sign. As a result, the Berry flux for valley K, $F^K = \Sigma_p f^K(p) \Omega^K(p)$, displays a rapid change as a function of chemical potential close to band edge; away from the band edge it saturates (see Fig. 3d). Here, $f^K(p)$ is the electronic distribution function for valley K. When both valleys are equally populated (valley symmetric), the net Berry flux $F_{total} = F^K + F^{K'}$ that tracks the strength of the AHE[19,20] is zero. Crucially, close to the band edge in the VM region, BTG possess $D$ tuned van Hove singularities with pronounced peaks in

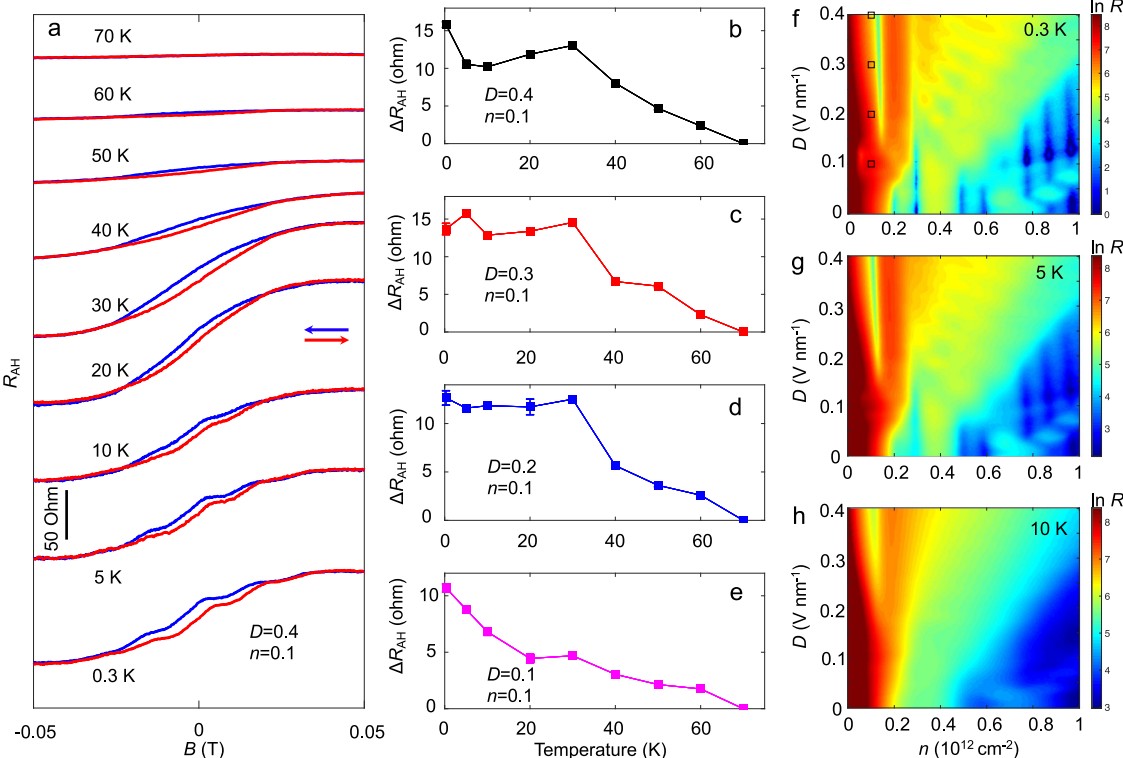

**Fig. 4 | Temperature dependence of AHE in BTG. a** Anomalous Hall resistance ($R_{AH}$) as a function of magnetic field at ($D = 0.4$, $n = 0.1$) at different temperatures. **b–e** Temperature dependence of the $\Delta R_{AH}$ with at different ($n$, $D$) conditions. Symbols for experimental data are shown with error bars (error bars are extracted from the noise level in data, some are smaller than symbols). **f–h** $R_{xx}$ as a function of

$n$ and $D$ at $B = 1$ T at different temperatures, 0.3, 5, and 10 K, respectively. $n$ and $D$ are in units of $10^{12}$ cm$^{-2}$ and V nm$^{-1}$, respectively. The ($n$, $D$) conditions of **a–e** are denoted in f with black hollow squares respectively; note that panel a and panel b are at the same ($n$, $D$) condition.

the density of states (Fig. 1b): these enhance interactions to facilitate ferromagnetic ordering and a Stoner-type lowering of the four-fold isospin degeneracy[6,16,17,21–24]. Van Hove driven symmetry breaking in two- and three-layer graphene have recently been found to yield a myriad of spin, valley, and other flavor polarized states[6,16,17,21–25]. In BTG, AHE in Fig. 3 corresponds to a spontaneous broken valley symmetry characterized by a spontaneous valley polarization: $f^K(\mathbf{p}) \neq f^{K'}(\mathbf{p})$. Spontaneous valley polarization features a non-zero $F^{\pm}_{total}$ for each magnetic state ($\pm$), manifesting as a hysteretic $R_{AH}$ and a non-zero $R_{xy}$ at $B = 0$, as shown in Fig. 3a, b.

Because electrons in each of the valleys possess contrasting magnetic moments[26], the valley polarization [with $f^K(\mathbf{p}) \neq f^{K'}(\mathbf{p})$] can be further tuned by $B$ field (via an orbital magnetic Zeeman-like coupling $\mathbf{m}^{K,K'} \cdot \mathbf{B}$ in each valley[18]) favouring one valley over the other, see also Fig. 1b. This manifests as a ramp-like profile of $R_{AH}$ with applied $B$. Importantly, because Berry curvature is concentrated at the band edge, $dF^{K,K'}/d\mu$ is most pronounced at the band edge (Fig. 3d). As a result, $B$ field induced changes to $F^{\pm}_{total}$, and concomitantly $R_{AH}$, are expected to rapidly saturate.

This VM picture of the AHE is consistent with the phenomenology of BTG AHE observed in our devices. For instance, the peaked nature of $dF^{K,K'}/d\mu$ also mean that both the size of the hysteretic window as well as the magnitude of saturated $R_{AH}$ should be most pronounced at low density. To see this, we plot $R_{AH}$ as a function of $B$ for decreasing $n$ in Fig. 3e. At moderate density $n = 1.0 \times 10^{12}$ cm$^{-2}$ (top curve), $R_{AH}$ is flat with no appreciable hysteresis. As density is lowered, both the size of the hysteretic window and the saturated $R_{AH}$ increase, in agreement with the peaked $dF^{K,K'}/d\mu$ close to the band edge. This density sensitivity allows BTG AHE to be switched "on" by tuning into the low electron density region where Berry flux changes rapidly. Similarly, since a finite $D$ field turns on Berry curvature in the valleys of BTG, we

find that AHE can also be switched "on" by $D$ field (see Supplementary Section 4 for details).

At the lowest density, additional steps in the $R_{AH}$ appear (Fig. 3b and Fig. 3e (bottom)), yielding a staircase-like profile as a function of $B$. These steps appear across successive scans of $B$ field (see Supplementary Fig. 13) and could arise from a number of mechanisms, including for example domain inhomogeneity with slightly different coercivities[3], spatial charge density inhomogeneity, or momentum polarized pockets in each valley defined by the trigonally warped band structure[23]. In the latter scenario, pocket polarization is turned on only at very low density; since each pocket contains a hot spot of Berry curvature (see Fig. 3c), sequential filling of these may lead to a staircase profile of $R_{AH}$. We note that momentum polarized pockets have recently been observed in Bernal stacked bilayer graphene[25]. Notwithstanding the fact that we have concentrated on electron doping in the main text, a hysteretic AHE also exists when BTG is doped to the hole side (Supplementary Section 5). This indicates that valley symmetry can also be broken for holes in BTG.

The temperature dependence of the anomalous Hall effect is displayed in Fig. 4. Anomalous Hall resistance (after subtraction of linear background) at $D = 0.4$ V nm$^{-1}$ and $n = 0.1 \times 10^{12}$ cm$^{-2}$ for different temperatures is shown in Fig. 4a. While the fine structure of AHE (e.g., steps) is washed out with increasing temperature, BTG AHE survives up to temperatures as large as several tens of kelvin, which is remarkable. To reveal the robustness of the high temperature phenomena, we show the difference of the Hall resistance between forward and backward sweeps at zero magnetic field, $\Delta R_{AH}$, as temperature is varied for distinct values of $D$ field at fixed carrier density of $n = 0.1 \times 10^{12}$ cm$^{-2}$ in Fig. 4b–e. We find non-zero $\Delta R_{AH}$ persists up to temperatures as large as several tens of kelvin for a variety of $D$ field values; this contrasts with AHE in rhombohedral trilayer graphene where AHE

disappears at a few kelvin[5,6]. We note that BTG quantum oscillations are washed out at around 10 kelvin (Fig. 4f–h), making AHE a thermally robust signature of the lowered degeneracy and broken symmetry in BTG. Results from additional devices with AHE are also presented in Supplementary Fig. 14.

It is interesting to compare BTG AHE discussed here with other AHE found in graphene stacks, e.g., in rhombohedral trilayer graphene[6] and twisted bilayer graphene aligned with hexagonal boron nitride[3,4]. The AHE in these systems occur when both valley and spin symmetries are simultaneously broken. In rhombohedral trilayer graphene, AHE only occurs in a quarter metal state and conspicuously vanishes in the half metallic state[6]. The vanishing of AHE in the half-metallic state is a signature of the spin-polarized (but valley unpolarized) nature of the half-metallicity in rhombohedral trilayer graphene[6]. This arises due to the minimal spin-orbit coupling in graphene multilayers that yields a Berry curvature which has the same sign for opposite spins. We note that evidence for spin polarized half-metallic states have also been reported more generally in rhombohedral stacked few-layer graphene (both rhombohedral trilayers and rhombohedral tetralayers)[27].

Similarly, in twisted bilayer graphene heterostructures, AHE occurs near 3/4 filling of a moiré flat band[3,4] where both spin- and valley- symmetry are broken. In sharp contrast, in BTG, we find that AHE occurs in the VM region where the frequency of quantum oscillations are consistent with a half metal while we found no signatures of a quarter metallic state. This indicates the valley polarized half metal origin of BTG AHE.

## Discussion

The key ingredients of BTG AHE are valley contrasting Berry curvature and the high density of states close to $D$ field tuned van Hove singularities, which are naturally found in gated (even) layers of Bernal-stacked graphene. As a result, we anticipate that AHE and spontaneously valley-polarized states are likely ubiquitous in gapped graphene multilayers. Therefore our results imply that the energetically *stable* form of graphene multilayers can be used to probe and exploit graphene based AHE beyond the metastable configurations of moiré superlattices[3,4] and rhombohedral trilayer graphene[5,6]. Perhaps the most exciting aspect is the ability to sustain valley polarization at zero magnetic field. Requiring both inversion symmetry breaking (controlled by $D$ field) as well as spontaneous time-reversal symmetry breaking, such valley polarization mirrors the requirements for magneto-chiral responses[28], which has implications for valley-mediated intrinsic non-reciprocity. Further, since BTG ferromagnetism is turned on by $D$ field, they are ideal candidates to be readily combined with layered (out-of-plane) ferroelectrics[29,30] to achieve stack- engineered multiferroic functionality.

## Methods
### Device fabrication
Tetralayer graphene, graphite and hexagonal boron nitride (hBN) flakes are exfoliated on silicon dioxide/silicon wafers. The whole heterostructure was assembled using dry-transfer technique using a PC (polycarbonate) film/PDMS stamp. We first pick up the top graphite and subsequently top hBN, BTG, bottom hBN and bottom graphite and put the whole stack on another wafer. The hBN layers are not intentionally aligned with BTG sample. Graphite gates are employed to reduce disorder and ensure high quality devices. Finally the device was etched into a Hall bar and contacted using edge-contact method with Cr (5 nm)/Au (70 nm).

### Transport measurements
Transport measurements were conducted in a variable-temperature He-3 or He-4 Probe in Oxford Teslatron system. The lowest temperature reached is ~300 mK. Standard lock-in technique is used to measure the four-probe voltages using SR830. Low pass filters are used to suppress low frequency noises. Constant current source is applied with ranges from 10 to 100 nA. Gate voltages are applied using Keithley 2400 and 6430. Error bars in determining $\Delta R_{AH}$ are limited by the electrical noise in data acquisition.

## Data availability
All data needed to evaluate the conclusions in the study are present in the paper and/or the Supplementary Information. All data that support the findings within this paper are available from the corresponding authors upon request.

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

## Acknowledgements

K.P.L. wishes to thank Singapore's National Research Foundation competitive research program grant NRF CRP22-2019-0006. J.C.W.S. acknowledges support by Singapore's Ministry of Education (MOE) Academic Research Fund Tier 3 Grant MOE2018-T3–1-002.

## Author contributions

H.C. and K.P.L. conceived the project. H.C. carried out device fabrications and transport measurements. A.A. and J.C.W.S. provided theoretical inputs. H.C., A.A., J.C.W.S. and K.P.L. wrote the manuscript together.

## Competing interests

The authors declare no competing interests.
