## [Peer Review File · Nature Communications]

REVIEWER COMMENTS

Reviewer #1 (Remarks to the Author):

In this work, the authors carried out magnetoresistance measurements in dual-gated Bernal-stacked tetralayer graphene (BTG). Using the dual-gated structure, authors tuned displacement field (D) and charge density (n) independently and observed anomalous Hall effect (AHE) at large displacement field (D) and low carrier density (n) in the region labelled as a valley metal ('VM'). This is striking because it suggests that the time-reversal symmetry is spontaneously broken at zero magnetic fields. Moreover, quantum oscillation analysis shows that the total degeneracy is two in the VM region (a half metal) instead of one (so-called a quarter metal), indicating that the observed AHE is different from those found in rhombohedral trilayer and twisted bilayer graphene aligned with hBN. To find out the origin and support their claim, authors calculated Berry curvature of the BTG band at finite D near zero density and showed that non-zero Berry flux exists near zero energy. However, since the Berry flux is opposite in different valleys, to explain their observation, the valley degeneracy has to be broken. The authors claim that this is from the strong interactions at the van Hove singularity point that presents near zero energy coexisting with a finite Berry flux.

The work is interesting and shows that Bernal-stacked graphene multilayers can exhibit many-body physics even without further engineering like twisting or changing stacking order. However, there seem some weak points that need to be addressed before being accepted for publication in Nature Communications. Here are my comments and questions:

1) In previous study on BTG (Ref. 15), Landau level sequence (or the lowest Landau level observed at the lowest magnetic fields) was also checked to confirm the stacking order and number of layers. Although Figs. 1c and 2a look quite close to those shown in Ref. 15, it would be good to show the quantum-Hall effect data in Supplementary Information or mark the corresponding Landau level filling factors for the vertical stripes in Fig. 2a such that the readers can check the Landau level sequence. In the current form, there seems no clear way for the readers to check the Landau level filling factors.

2) According to the calculation shown in Fig. 1b and Supplementary Fig. 6b, the hole band seems to be flatter with a larger density of states. However, in the analysis, authors didn't present the corresponding data nor discuss it. In fact, in Fig. 2a, the hole side does not exhibit a clear stich-like ripple pattern (i.e., the signature of the VM region). It would be interesting to see if this contrast is intrinsic or comes from the sample quality. For instance, since the band dispersion is different on the hole side, Berry flux may remain zero nor valley degeneracy is not broken.

3) Authors claim that in FM region, AHE disappears and shows some AHE data measured at different D and n . However, it is not straightforward to check the crossing points, for instance, in Fig. 2a. It would be good to mark the positions in (n,D) at which the AHE disappears in Fig. 2a or somewhere the authors prefer. Alternatively, authors can add the points in Fig. 1c as it is measured at zero magnetic fields where AHE appears.

4) Related to the question #3, it seems that at $n=0.1$, where the authors present most of the AHE data, the longitudinal resistance measured at finite magnetic fields (see, e.g., Figs. 2a and 4f) shows a clear dip accompanied by more than two orders of magnitude larger resistance nearby. What is the origin of this state? Is it confirmed that this state is 'VM'?

5) In the band structure calculation, the authors assume that the two graphene layers in the middle (Supplementary equation 4) remain neutral and use Ref. 5 to support this assumption. However, Ref. 5 is for a trilayer graphene such that it is general to set $(V/2, 0, -V/2)$ from top to the bottom layers by assuming a homogeneous electric field distribution across the layers. In fact, for four-layers, generally, $(3V/2, V/2, -V/2, -3V/2)$ or $(V/2, V/6, -V/6, -V/2)$ is used. Especially near zero density, electric field cannot be screened so perfectly that only the top-most and bottom-most layers feel the potential V from the gate (you can check few literatures like PRB 102 035421). Authors either need to calculate the band structure and possibly also the Berry curvature again or provide stronger arguments why the Supplementary equation (4) can be used.

6) To help readers quickly find the relevant figures in the SI, please specify the figure number when referring to them in the main text.

7) There are few typos and mistakes in the paper. Please check the manuscript and supplementary carefully again. Here are few that I found: i) At page 2: there is an extra minus sign in 'The carrier density $n = [C_t(V_t - V_{t0}) + C_b(V_b - V_{b0})]/e'$, ii) At page 4: authors wrote 'Total Hall resistance (without subtraction of linear background) at $D=0.4$ V/nm ...' when they refer to Fig. 4a but the figure shows RAH, iii) the information about the magnetic fields seem missing in Figs. 4f-h, iv) The reference 19 has two papers.

Reviewer #2 (Remarks to the Author):

Reviewer #3 (Remarks to the Author):

The authors report an anomalous Hall effect (AHE) and magnetic hysteresis in Bernal tetralayer graphene. The data are high quality and the flow of the text and claims are clear. While high-quality samples of Bernal tetralayer graphene have been explored with low-temperature transport before (e.g. Refs. 15,16), the authors here focus on the behavior of their samples at low carrier densities, with finite electric displacement field, and low/zero magnetic field. In that regime, they find evidence of spin/valley flavor-symmetry breaking and the primary result, anomalous Hall resistance and hysteresis.

They ascribe the emergence of anomalous Hall resistance to net orbital magnetism developing within a "valley metal" region of the low-temperature phase diagram. Interestingly, they observe that the AHE persists to several tens of kelvin, a surprisingly high temperature given the weak magnetic hysteresis observed at the lowest temperatures, and given similar anomalous behavior observed in rhombohedral trilayer graphene only at lower temperatures. Another intriguing result is the development of a series of plateaux in the Hall resistance as a function of magnetic field, though the authors do not comment extensively on the latter.

Overall, their argument for a net valley imbalance and orbital magnetism is convincing, aside from some details that I will outline below. The results are timely and relevant to the discussion of related effects in moire and non-moire graphene systems such as Bernal and rhombohedrally stacked multilayer graphene. After addressing the following points, I believe the manuscript will be suitable for publication in Nature Communications.

1. The authors focus on the behavior of electrons in the manuscript, but they do not address whether they observed any anomalous Hall signatures for holes. Do they only see a linear Hall component at low densities and large displacement fields for holes? According to their DOS calculation (Fig. 1b), the Van Hove singularities in the valence band appear at least as large if not larger than in the conduction band. Naively, one might therefore expect symmetry breaking for holes as well as electrons (and possibly an AHE depending on the flavor polarization). Do they see evidence of symmetry breaking in the quantum oscillations for holes?

2. The boundary between the full and valley metals shown in Fig. 2a (gray line) appears quite sharp, but the onset of anomalous Hall resistance and hysteresis is gradual as a function of density near that boundary (e.g. Fig. 3e). Do the authors have a simple explanation for that difference? Does the density- or energy-dependence of the calculated Berry curvature account for the smooth onset of anomalous Hall?

3. Along those lines, what is the lower density limit for the observed AHE? Does the hysteresis loop close suddenly as the chemical potential approaches the gap or is there a gradual decrease in the AHE at densities below $n=0.1$?

4. The step-like features observed in R_{xy} at low temperatures are intriguing, but to rule out possible domain physics, did the authors measure similar steps in additional devices? Supplementary Fig. 9 does not seem to show such steps. In the primary device shown in the main text, did the authors check if the steps are consistent between multiple contact pairs? If they result from spatial inhomogeneity rather than something more exotic, like momentum polarization, different contact pairs may show steps occurring at different coercive fields.

5. Do the authors have a map of R_{xy} measured at $B=0$? This may help identify at a glance where the AHE is most pronounced.

Minor corrections:

6. In Fig. 1b, it would be informative to include real DOS units rather than arbitrary units. This will aid in comparing the magnitude of the Van Hove singularities to other related systems (Bernal, rhombohedral, moire, etc).

7. The color scale and/or color map used in Fig. 1a should be selected to show the quantum oscillations in the low-density ("VM") regime more clearly since this region is the focus of the manuscript.

8. The curves in Fig. 2b are not labeled according to where they were extracted from Fig. 2a (the caption mentions "corresponding symbols" but does not show them).

9. It would be much easier to understand the AHE behavior shown in Fig. 3 if there were an inset of the map shown in Fig. 2a with corresponding points or lines superimposed to show the location of each hysteresis loop. An alternative would be to add matching labels in Fig. 2a, but this may appear quite busy since there are already labeled points in that map for the FFT analysis.

10. I suggest doing something similar in Fig. 4, adding labels for the curves shown in Fig. 4a-e in the map shown in Fig. 4f.

11. In the final paragraph, the authors mention "magnetic field-free addressability of the valleys" in BTG. In what sense are the valleys addressable at zero magnetic field? Unless the authors have evidence of magnetic switching with electric control (e.g. current, density, or displacement field switching), it seems that a magnetic field is required to switch and address different valley-polarized states.

12. Ref. 9 seems out of place in the abstract sentence, "Here we report an anomalous Hall effect in Bernal-stacked tetralayer graphene devices (BTG), the most stable configuration of four layer graphene [7-9]." In addition, it would make sense to cite Refs. 15,16 in that sentence.

13. Refs. 19,20 should be added to the sentences, "...these enhance interactions to facilitate ferromagnetic ordering and a Stoner-type lowering of the four-fold isospin degeneracy [6,22-25]. Van Hove driven symmetry breaking in two and three layer graphene have recently been found to yield a myriad of spin, valley, and other flavor polarized states [6,22-26]."

Reviewer #4 (Remarks to the Author):

Reviewer #5 (Remarks to the Author):

The authors of this work demonstrate the existence of an anomalous hall effect in tetralayer graphene in the presence

of a finite magnetic field and finite displacement field. The result is important and the measurements seem reliable. I would like the authors to reply to the following questions before allowing for publication of the paper.

1) The authors should rationalize better the reason why the AHE occurs only at low carriers concentration

and high-displacement field. In particular, I am not so convinced by the need

of an high displacement field. Is it due to the experimental difficulties in detecting the effect or is there any conceptual reason for which we should have a large enough displacement field ?

Can the authors explain this better by looking at previous theoretical works on the subject or by using

their 8 bands TB model ?

2) An half metallic state associated with a broken symmetry state breaking time reversal was also detected recently in few layers rhombohedral graphene (see for example <https://doi.org/10.1021/acs.nanolett.2c00466>). In that case it was attributed to a layer antiferromagnetic

state. Could it be that some kind of magnetic state is present also here (with a weak magnetization) ? Can the author exclude in some way that the broken symmetry state is magnetic ?

Point-by-Point Responses

Reviewer #1 (Remarks to the Author):

In this work, the authors carried out magnetoresistance measurements in dual-gated Bernal-stacked tetralayer graphene (BTG). Using the dual-gated structure, authors tuned displacement field (D) and charge density (n) independently and observed anomalous Hall effect (AHE) at large displacement field (D) and low carrier density (n) in the region labelled as a valley metal ('VM'). This is striking because it suggests that the time-reversal symmetry is spontaneously broken at zero magnetic fields. Moreover, quantum oscillation analysis shows that the total degeneracy is two in the VM region (a half metal) instead of one (so-called a quarter metal), indicating that the observed AHE is different from those found in rhombohedral trilayer and twisted bilayer graphene aligned with hBN. To find out the origin and support their claim, authors calculated Berry curvature of the BTG band at finite D near zero density and showed that non-zero Berry flux exists near zero energy. However, since the Berry flux is opposite in different valleys, to explain their observation, the valley degeneracy has to be broken. The authors claim that this is from the strong interactions at the van Hove singularity point that presents near zero energy coexisting with a finite Berry flux.

The work is interesting and shows that Bernal-stacked graphene multilayers can exhibit many-body physics even without further engineering like twisting or changing stacking order. However, there seem some weak points that need to be addressed before being accepted for publication in Nature Communications. Here are my comments and questions:

Response:

We thank the reviewer for recognizing the novelty and value of our work. We also thank the reviewer for providing constructive comments and suggestions which we have fully adopted and addressed. We have prepared a point-by-point response to these comments below including appropriate changes in the manuscript and supplementary information. All changes have been highlighted in yellow. We believe these changes significantly improve the quality of our work and address the questions of the reviewer.

1) In previous study on BTG (Ref. 15), Landau level sequence (or the lowest Landau level observed at the lowest magnetic fields) was also checked to confirm the stacking order and number of layers. Although Figs. 1c and 2a look quite close to those shown in Ref. 15, it would be good to show the quantum-Hall effect data in Supplementary Information or mark the corresponding Landau level filling factors for the vertical stripes in Fig. 2a such that the readers can check the Landau level sequence. In the current form, there seems no clear way for the readers to check the Landau level filling factors.

Response

We thank reviewer for this constructive suggestion. As suggested by reviewer and to make reading the Landau levels clearer, we have re-plotted the Fig. 2a with filling factors ν labelled in the up x-axis (shown below as Figure R1 for the reviewer's convenience). Here $\nu = nh/eB$, where n , h , e and B denotes carrier density, Planck constant, elementary charge and magnetic field, respectively. This replotted Fig. R1 demonstrates the Landau level sequences clearly. Additionally, to avoid an over-crowded Fig. 2a, we have directly placed this replotted figure in the Supplementary Information as suggested by the reviewer as Supplementary Fig. 2; please see also updated SI. We have also added a sentence in the main text that points the reader to Supplementary Fig. 2 to see the Landau level sequence.

Fig. R1. Carrier density and displacement field dependence of four probe resistance under $B=1\text{T}$. The lower x-axis labels carrier density and the upper x-axis label display the filling factors $\nu=nh/eB$, respectively. Here we have concentrated on the electron side where the oscillations are most pronounced.

2) According to the calculation shown in Fig. 1b and Supplementary Fig. 6b, the hole band seems to be flatter with a larger density of states. However, in the analysis, authors didn't present the corresponding data nor discuss it. In fact, in Fig. 2a, the hole side does not exhibit a clear stitch-like ripple pattern (i.e., the signature of the VM region). It would be interesting to see if this contrast is intrinsic or comes from the sample quality. For instance, since the band dispersion is different on the hole side, Berry flux may remain zero nor valley degeneracy is not broken.

Response:

We thank the reviewer for pointing out the faint stitch-like ripple pattern on hole side.

We agree that the resistance plot is not very well resolved on the hole side. This lower resolution on the hole side was also observed in previous experiments on Bernal tetralayer graphene, e.g., figure 4a in Y. Shi, et al. Phys Rev Lett 120, 096802. Of course, we agree with the reviewer that the hole side can also manifest regions of large density of states and energy windows with flat-like bandstructure. This clearly motivates the reviewer's interesting question about whether spontaneous Valley Symmetry breaking also manifests in the hole side. Indeed (as anticipated by the reviewer), we find that a hysteretic AHE emerges on the hole side, see Figure R2 plotted below for the convenience of the reviewer. A hysteretic AHE is a clear tell-tale signature of the broken valley symmetry in BTG which, as we now see, also appears in both electron and hole sides. Figure R2 and a short description can be now found in Supplementary Fig. 6 and Supplementary Section 5, respectively. We have also modified the main text to describe that hysteretic AHE signatures persist in the hole side as well, and have pointed to Supplementary Section 5 where the AHE data for the hole side is presented.

Similar to electron side, the hysteretic AHE curve in the hole side becomes more pronounced at lower hole density (see Fig. R2a-d); AHE seems to vanish for hole densities beyond 10^{12} cm^{-2} hole. While the hole side does exhibit some knee-like step features (Figure R2a) they are less pronounced than what we observed for the electron side (Figure 3).

Fig. R2. Hysteresis and anomalous Hall curves of hole side, with different carrier density conditions at fixed $D=0.3 \text{ V nm}^{-1}$. n and D are in units of 10^{12} cm^{-2} and V nm^{-1} , respectively. Here the negative n values indicate hole density.

3) Authors claim that in FM region, AHE disappears and shows some AHE data measured at different D and n . However, it is not straightforward to check the crossing points, for instance, in Fig. 2a. It would be good to mark the positions in (n, D) at which the AHE disappears in Fig. 2a or somewhere the authors prefer. Alternatively, authors can add the points in Fig. 1c as it is measured at zero magnetic fields where AHE appears.

Response:

We thank the reviewer for pointing this out and for the excellent suggestion to improve the readability of the manuscript. As suggested by the reviewer, we have now added add black cross symbols ('X') directly in Fig. 1c to mark the (n, D) conditions corresponding to those in Fig. 3e. As can be seen, the AHE (almost) disappears in the top trace of Fig. 3e (corresponding to $n=1.0$ and $D=0.3$). This $(n=1.0, D=0.3)$ condition corresponds to the right most cross symbol (X) in Fig. 1c. All labels have been described in both the captions of Figure 1 as well as Fig. 3.

4) Related to the question #3, it seems that at $n=0.1$, where the authors present most of the AHE data, the longitudinal resistance measured at finite magnetic fields (see, e.g., Figs. 2a and 4f) shows a clear dip accompanied by more than two orders of magnitude larger resistance nearby. What is the origin of this state? Is it confirmed that this state is 'VM'?

Response:

We thank the reviewer for pointing out this issue. To clarify, the color bar in Fig. 2a and Fig. 4f are in ln-scale (natural logarithm, with base $e \sim 2.718$) instead of log-scale (with base 10), and therefore the BTG does not have a drastic resistance change near the $n=0.1$ region. Instead, the change in resistance is e^2 or by a factor of about 7. This convention is indicated by the label "ln R_{xx} " in Figure 1 and 2 of the main text.

Importantly, this resistance change does not occur at $B=0$ (see Fig. 1c of main text); it only occurs at finite B . To draw conclusions about the VM state, it is important to study the $B=0$ behavior.

Importantly, in our work we have concentrated on valley symmetry breaking and the reporting the observation of anomalous Hall effect in BTG (*i.e.* at $B=0$). In inversion broken BTG, where the valleys have opposite signs of Berry curvature, a hysteretic anomalous Hall effect is a tell-tale signature of valley symmetry breaking. Indeed, in the $n=0.1$ region, we find that the AHE in BTG becomes more pronounced as compared with higher densities, see e.g., Fig. 3e of the main text. This is a clear signature of spontaneous valley symmetry breaking. Further, the relatively low resistance of this state ~ 200 - 300 ohms at zero magnetic field indicates a metallic character of the transport in BTG in this region. As such, we can confirm that the state around the $n=0.1$ region is a spontaneous valley symmetry broken state – *i.e.*, VM state.

5) In the band structure calculation, the authors assume that the two graphene layers in the middle (Supplementary equation 4) remain neutral and use Ref. 5 to support this assumption. However, Ref. 5 is for a trilayer graphene such that it is general to set $(V/2, 0, -V/2)$ from top to the bottom layers by assuming a homogeneous electric field distribution across the layers. In fact, for four-layers, generally, $(3V/2, V/2, -V/2, -3V/2)$ or $(V/2, V/6, -V/6, -V/2)$ is used. Especially near zero density, electric field cannot be screened so perfectly that only the top-most and bottom-most layers feel the potential V from the gate (you can check few literatures like PRB 102 035421). Authors either need to calculate the band structure and possibly also the Berry curvature again or provide stronger arguments why the Supplementary equation (4) can be used.

Response: We thank the reviewer for pointing out how elsewhere in the literature [e.g., Phys Rev B 102 035421 (2023)], a different form of the potential drop in the middle layers of BTG has been used. We had originally used the simplified model in our Supplementary equation 4 as a simplified illustration of the BTG bandstructure taking into account for layer screening effects sometimes discussed in the literature for multilayer graphene (see e.g., Phys Rev B 81, 125304 [2010]). Of course, the reviewer's point is well taken: the screening in the middle layers is likely not as strong as the simple Supplementary equation 4 suggests. While there is some screening, it is likely that the situation is likely in between the equal potential drop scenario that the reviewer suggests and the "toy" unequal potential drop scenario used in Supplementary equation 4. Nevertheless, we expect that the qualitative conclusions of our model should persist even for the reviewer's suggested potential drop scenario.

To see this explicitly, in the new Supplementary Section 9, we have now used the form of potential across the 4 layers that the reviewer suggests. We have re-simulated the bandstructure and berry curvature using this new potential. For clarity, we have chosen potential $(\Delta, \Delta/3, -\Delta/3, -\Delta)$ on the four layers to maintain a total potential difference of 2Δ between the outermost layers for ease of comparison with our Supplementary Equation 4; this form is equivalent to the form that the reviewer suggests (it is just scaled so that outermost layers are Δ and $-\Delta$ respectively). Importantly, we find that the qualitative features of the band structure and the berry curvature in Supplementary Section 9 are similar to that simulated using our Supplementary Equation 4, namely: displacement field induced Berry curvature are concentrated around pockets close to the band bottom, a band gap is opened up when a displacement field is applied (see Supplementary Fig. 12 for new bandstructure and berry curvature distributions), and Berry curvature changes sign for opposite valleys. Of course, there are quantitative changes, e.g., the berry curvature is more concentrated. Nevertheless, the broad qualitative features are not affected: a hysteretic anomalous Hall effect arises from broken valley symmetry and is most pronounced for carrier densities close to band bottom where berry curvature distribution is most pronounced.

For clarity to the reader, we now display **both models**. We have retained our model in Supplementary Section 6 as well as in the main text and at the same time show the simulations from the BTG model using $(\Delta, \Delta/3, -\Delta/3, -\Delta)$ [as suggested by the reviewer] in Supplementary Section 9. Both in the main text (see captions of the Figs. 1 and 3) and in the Supplementary Information, we have made reference to both models pointing to the Supplementary Information.

6) To help readers quickly find the relevant figures in the SI, please specify the figure number when referring to them in the main text.

Response:

We thank the reviewer for the kind suggestion. We have implemented the corresponding changes to enhance the readability of our text, especially when making reference to figures in the Supplementary Information.

7) There are few typos and mistakes in the paper. Please check the manuscript and supplementary carefully again. Here are few that I found: i) At page 2: there is an extra minus sign in 'The carrier density $n=[C_t(V_t-V_{t0})+C_b(V_b-V_{b0})]/e$ ', ii) At page 4: authors wrote 'Total Hall resistance (without subtraction of linear background) at $D=0.4$ V/nm ...' when they refer to Fig. 4a but the figure shows RAH, iii) the information about the magnetic fields seem missing in Figs. 4f-h, iv) The reference 19 has two papers.

Response: We are sorry for these typos and errors and we would like to thank reviewer for carefully reading our manuscript and pointing these out. As suggested, we have rectified these typos and errors. i) The extra minus sign is deleted; ii) The 'Total Hall resistance (without subtraction of linear background) at $D=0.4$ V/nm ...' has been changed to 'Anomalous Hall resistance (after subtraction of linear background)'. iii) The magnetic fields condition (*i.e.*, $B=1$ T) for Figs. 4f-h have been added into the caption. iv) We have corrected reference 19 [the second reference in the two displayed articles has been removed; it was a repeated reference that was already cited elsewhere]. Thanks again for the careful reviewing.

Reviewer #2 (Remarks to the Author):

Response: We sincerely thank the reviewer for reviewing our manuscript and for providing constructive suggestions. We have carefully revised the manuscript, supplementary information and reply to the concerns and questions to all reviewers in this response letter (see point-by-point replies). We believe this revision significantly improve the quality of our work.

Reviewer #3 (Remarks to the Author):

The authors report an anomalous Hall effect (AHE) and magnetic hysteresis in Bernal tetralayer graphene. The data are high quality and the flow of the text and claims are clear. While high-quality samples of Bernal tetralayer graphene have been explored with low-temperature transport before (e.g. Refs. 15,16), the authors here focus on the behavior of their samples at low carrier densities, with finite electric displacement field, and low/zero magnetic field. In that regime, they find evidence of spin/valley flavor-symmetry breaking and the primary result, anomalous Hall resistance and hysteresis.

They ascribe the emergence of anomalous Hall resistance to net orbital magnetism developing within a "valley metal" region of the low-temperature phase diagram. Interestingly, they observe that the AHE persists to several tens of kelvin, a surprisingly high temperature given the weak magnetic hysteresis observed at the lowest temperatures, and given similar anomalous behavior observed in rhombohedral trilayer graphene only at lower temperatures. Another intriguing result is the development of a series of plateaux in the Hall resistance as a function of magnetic field, though the authors do not comment extensively on the latter.

Overall, their argument for a net valley imbalance and orbital magnetism is convincing, aside from some details that I will outline below. The results are timely and relevant to the discussion of related effects in moire and non-moire graphene systems such as Bernal and rhombohedrally stacked multilayer graphene. After addressing the following points, I believe the manuscript will be suitable for publication in Nature Communications.

Response:

We thank the reviewer for the positive and valuable comments for our manuscript. We have addressed all comments and adopted all the changes suggested by the reviewer. Additionally, we have prepared a point-by-point response to these comments below including appropriate changes in the manuscript and supplementary information. All changes have been highlighted in yellow. We believe these changes significantly improve the quality of our work and address the questions of the reviewer.

1. The authors focus on the behavior of electrons in the manuscript, but they do not address whether they observed any anomalous Hall signatures for holes. Do they only see a linear Hall component at low densities and large displacement fields for holes? According to their DOS calculation (Fig. 1b), the Van Hove singularities in the valence band appear at least as large if not larger than in the conduction band. Naively, one might therefore expect symmetry breaking for holes as well as electrons (and possibly an AHE depending on the flavor polarization). Do they see evidence of symmetry breaking in the quantum oscillations for holes?

Response: We thank the reviewer for the excellent question. For clarity, we will structure our reply in two parts:

1. While we presented quantum oscillation analysis, the key tell-tale signature of **spontaneous** symmetry breaking (in our case, valley symmetry breaking) is that of the anomalous Hall effect (i.e. a Hall effect at $B=0$). The hysteresis curves we observed indicate that time-reversal symmetry is broken and importantly (since Berry curvature has opposite signs for opposite valleys) valley symmetry is broken.
2. The reviewer is right that the density of states in the hole band can be large and that this may also lead to spontaneous symmetry breaking much like on the electron side. To see this we checked the hole side for signatures of an anomalous hall effect at $B=0$. As anticipated by the reviewer, we found an anomalous Hall effect on the hole side. This is shown below for the convenience of reviewer as Fig. R3, it is also now described in the revised supplementary information as Supplementary Section 5 and Supplementary Fig. 6. This indeed shows that spontaneous Valley symmetry breaking also manifests on the hole side.

Fig. R3. Hysteresis and anomalous Hall curves of hole side, with different carrier density conditions at fixed $D=0.3 \text{ V nm}^{-1}$. n and D are in units of 10^{12} cm^{-2} and V nm^{-1} , respectively. Here the negative n values indicate hole density.

Figure R3 and a short description can be now found in Supplementary Section 5 and Supplementary Fig. 6 and. We have also modified the main text to describe that hysteretic AHE signatures persist in the hole side as well, and have pointed to Supplementary Section 5 where the AHE data for the hole side is presented.

2. The boundary between the full and valley metals shown in Fig. 2a (gray line) appears quite sharp, but the onset of anomalous Hall resistance and hysteresis is gradual as a function of density near that boundary (e.g. Fig. 3e). Do they authors have a simple explanation for that difference? Does the density- or energy-dependence of the calculated Berry curvature account for the smooth onset of anomalous Hall?

Response: We thank the reviewer for the comment. The grey line was meant to be a “guide to the eye” visually separating out the VM and FM regions; it was not meant to indicate a sharp transition from the VM to the FM regions. Indeed, as the reviewer has already noticed, the onset of the anomalous Hall resistance and hysteresis in Figure 3 is gradual. We have now completely removed this grey line which is confusing. The reviewer is correct when noting that the smooth change of the Berry flux (see e.g., Fig. 3d) is consistent with the smooth onset of the anomalous Hall effect in our samples.

3. Along those lines, what is the lower density limit for the observed AHE? Does the hysteresis loop close suddenly as the chemical potential approaches the gap or is there a gradual decrease in the AHE at densities below $n=0.1$?

Response: We thank the reviewer for this excellent question. It is difficult to systematically determine from our experiments if there is a low density bound for the manifestation of AHE in BTG.

To see this from our data, we display our Hall effect data for lower densities specifically: $n = 0.08, 0.03$ and 0.01 ; these densities are normalized to 10^{12} cm^{-2} plotted as Fig. R4 below for the convenience of the reviewer. We found that AHE persists for densities below $n=0.1$ (namely $n=0.08$ and $n=0.03$, see Fig. R4 panel a and b). However, when we pushed to even lower densities (namely $n=0.01$, see Fig R4 panel c and d), the Hall effect becomes large and does not have the same linear in B background (at higher B

fields) as before; it deviates significantly from the form $R_{xy} = \eta B + R_{AH}$ that the Hall effect at higher densities seem to follow which makes it difficult for R_{AH} to be disentangled from the Hall signal. Instead, large hysteretic curves in R_{xy} with a complex form can be observed (see Fig. R4 panel c and d). These may arise from a number of sources, e.g., from magnetic domains, from electron-hole puddles that become polarized (valley polarized and/or momentum polarized), disorder, or a combination of all the above. Indeed, we note that even in high mobility graphene heterostructures, electron-hole puddles are of the order $\delta n \sim \text{several} \times 10^{12} \text{ cm}^{-2}$ [J. Martin, et al. Nature physics, 4(2), 144-148 (2008); A. Deshpande, et al. Phys Rev B 79.20: 205411 (2009)]. This prevents us from making systematic statements about AHE at extremely low doping.

We have now included a short discussion of this data at low density in Supplementary Section 6 and Supplementary Fig. 7.

Fig. R4. a,b, Anomalous Hall (R_{AH}) curves at two very low carrier densities. c,d, Hall (R_{xy}) curves at extremely low carrier density for two different Hall contacts as denoted in inset. n and D are in units of 10^{12} cm^{-2} and V nm^{-1} , respectively.

4. The step-like features observed in R_{xy} at low temperatures are intriguing, but to rule out possible domain physics, did the authors measure similar steps in additional devices? Supplementary Fig. 9 does not seem to show such steps. In the primary device shown in the main text, did the authors check if the steps are consistent

between multiple contact pairs? If they result from spatial inhomogeneity rather than something more exotic, like momentum polarization, different contact pairs may show steps occurring at different coercive fields.

Response: We thank the reviewer for the excellent question and suggestions. For clarity, we answer in two parts:

1. In our manuscript, we tried to be agnostic to the specific mechanism of the steps. Instead, we presented all possibilities for the steps (see our main text) that include for instance magnetic domain physics as well as momentum polarization physics.
2. To the reviewer's question: we have additional data for the different contacts, see below in Figure R5. As can be seen, all curves feature step like features but occur at slightly different coercive fields or show different values of the anomalous Hall effect in each step. On face value, this may be indicative of magnetic domain-like physics. However, due to the low density in which this occurs, it could also equally arise from spatial inhomogeneities in the density as well as a possible interplay with a spatially non-uniform momentum polarization. It is difficult to rule out any of the above scenarios. We would like to emphasize that the main result of our work was a report of the observation of the AHE in BTG in general (see Figure 3 and Figure 4 of the main text). To more fully disentangle this physics, future local probe measurements coupled with transport may enable to disentangle the origin of the steps we have observed.

Figure R5. a, b, c, The anomalous Hall resistance as a function of magnetic field measured in three different Hall contacts in device BTG_1. All are acquired under same (n, D) condition as denoted in the plots.

5. Do the authors have a map of R_{xy} measured at $B=0$? This may help identify at a glance where the AHE is most pronounced.

Response: We thank the reviewer for the suggestion. While R_{xy} map as function of n and D at $B=0$ T could show where the nonzero R_{xy} value resides at zero magnetic field, the values R_{xy} alone does not indicate a hysteretic AHE behaviour. Instead, to systematically demonstrate the AHE, one would need to do forward and backward sweeps of B-field dependence at each fixed (n, D) conditions. We have now implemented this. As shown in Fig. R6, by implementing forward and backward sweeps in B-field and measuring R_{xy} , we extract the ΔR_{AH} values for the difference between back and forth curves of R_{xy} in B field scan and plot it as function of (n, D) states. As shown in the map, we can see that regions with finite ΔR_{AH} values are consistent with the region of VM states in Fig. 2 in the main text.

We have now included this colorplot figure as Supplementary Fig. 8.

Fig. R6. ΔR_{AH} as function of n and D conditions at $B=0$ T at 0.3 K.

Minor corrections:

6. In Fig. 1b, it would be informative to include real DOS units rather than arbitrary units. This will aid in comparing the magnitude of the Van Hove singularities to other related

systems (Bernal, rhombohedral, moire, etc).

Response: We thank the reviewer for this pointing out. Indeed, it would be helpful to have real DOS units in order to make comparisons. Therefore, following reviewer's suggestion we have included the units of DOS in main text SI. We now use $\text{eV}^{-1}A_{\text{uc}}^{-1}$ as the units for DOS where A_{uc} is the area of the unit cell.

7. The color scale and/or color map used in Fig. 1a should be selected to show the quantum oscillations in the low-density ("VM") regime more clearly since this region is the focus of the manuscript.

Response: We thank the reviewer for pointing this out. We assume there is a typo here and reviewer refers to Fig. 2a. Yes, we have tried to adjust the scale and colorbar in the (n, D) map but it is difficult to resolve the period of the quantum oscillations clearly in the VM region. Note that the VM region is filled with stitch-like patterns (consistent Physical Review Letters 120 , 096802 [2018] our Ref. 12 of the main text); the tilted stitch-like patterns stem from a complex pattern of Landau levels and their intersections that disperse with D . As a result, unlike the FM region where $\nu=4$ spacings are clearly visible, the tilted stitch patterns make it challenging to see spacings in the VM region. Instead, the period of quantum oscillations is most readily obtained by studying the B-field dependence of the R_{xx} (see e.g., Supplementary Fig. 3) and extracting its frequency (as shown in the main text Figure 2b).

8. The curves in Fig. 2b are not labeled according to where they were extracted from Fig. 2a (the caption mentions "corresponding symbols" but does not show them).

Response: We thank the reviewer for pointing this out. For clarity, the shapes of labels in Fig. 2b were drawn (from top to bottom curves are up-triangle, down-triangle, star, square, diamond, and circle respectively) in accordance with their (n, D) positions in Fig. 2a. We realize their sizes may have been a little small in the previous version making their shapes hard to see. To make these clearer for the reader, in the revised manuscript, we have enlarged these data symbols in Fig. 2b as well as stated their shapes explicitly in the caption of Figure 2 of the main text.

9. It would be much easier to understand the AHE behavior shown in Fig. 3 if there were an inset of the map shown in Fig. 2a with corresponding points or lines superimposed to show the location of each hysteresis loop. An alternative would be to add matching labels in Fig. 2a, but this may appear quite busy since there are already labeled points in that map for the FFT analysis.

Response: We thank the reviewer for the excellent suggestion. This suggestion was also raised up by reviewer 1 in question 3. We agree with the reviewer that adding labels to longitudinal resistance plots will make it easier to track the AHE behaviour. While we tried to add more labels to Fig. 2a, it makes the figure quite crowded. Instead, we have followed reviewer 1's suggestion: we have added black crosses (X) in Fig. 1c to mark the (n, D) conditions corresponding to those in Fig. 3e at $B=0$.

10. I suggest doing something similar in Fig. 4, adding labels for the curves shown in Fig. 4a-e in the map shown in Fig. 4f.

Response: We thank the reviewer for careful checking the manuscript. As suggested, we have added symbols (black hollow squares) in Fig. 4f at corresponding (n, D) conditions for convenient cross-checking of (n, D) conditions with Fig. 4a-e in the revised manuscript. Note that panel a and panel b are at the same (n, D) condition

11. In the final paragraph, the authors mention "magnetic field-free addressability of the valleys" in in BTG. In what sense are the valleys addressable at zero magnetic field? Unless the authors have evidence of magnetic switching with electric control (e.g. current, density, or displacement field switching), it seems that a magnetic field is required to switch and address different valley-polarized states.

Response: We thank the reviewer for the careful reading of our text. Indeed, we agree with the reviewer regarding the context of the word "addressability". All we wanted to say here is that we do not need a magnetic field to sustain the valley polarization. The valley polarization persists even at zero magnetic field. For clarity, we have now changed the sentence to say plainly what is seen: i.e. "sustain valley polarization at zero magnetic field".

12. Ref. 9 seems out of place in the abstract sentence, "Here we report an anomalous Hall effect in Bernal-stacked tetralayer graphene devices (BTG), the most stable configuration of four layer graphene [7-9]." In addition, it would make sense to cite Refs. 15,16 in that sentence.

Response: We thank the reviewer for careful reading. As suggested by reviewer, we have removed Ref. 9 and included Ref. 15 and 16 (now our Ref. 12 and 13 of the revised main text).

13. Refs. 19,20 should be added to the sentences, "...these enhance interactions to facilitate ferromagnetic ordering and a Stoner-type lowering of the four-fold isospin degeneracy [6,22-25]. Van Hove driven symmetry breaking in two and three layer graphene have recently been found to yield a myriad of spin, valley, and other flavor polarized states [6,22-26]."

Response: We thank the reviewer for the careful review of our text. We agree that refs 19 and 20 should be included as they address on the isospin broken states. They have been added accordingly.

Reviewer #4 (Remarks to the Author):

We sincerely thank the reviewer for reviewing our manuscript and for providing constructive suggestions. We have carefully revised the manuscript, supplementary information and reply to the concerns and questions to all reviewers in this response letter (see point-by-point replies). We believe this revision significantly improve the quality of our work.

Reviewer #5 (Remarks to the Author):

The authors of this work demonstrate the existence of an anomalous hall effect in tetralayer graphene in the presence of a finite magnetic field and finite displacement field. The result is important and the measurements seem reliable. I would like the authors to reply to the following questions before allowing for publication of the paper.

We thank the reviewer for the positive and valuable comments for our manuscript. We have addressed all comments and adopted all the changes suggested by the reviewer. Additionally, we have prepared a point-by-point response to these comments below including appropriate changes in the main text. All changes have been highlighted in yellow. We believe these changes significantly improve the quality of our work and address the questions of the reviewer.

1) The authors should rationalize better the reason why the AHE occurs only at low carriers concentration and high-displacement field. In particular, I am not so convinced by the need of an high displacement field. Is it due to the experimental difficulties in detecting the effect or is there any conceptual reason for which we should have a large enough displacement field ? Can the authors explain this better by looking at previous theoretical works on the subject or by using their 8 bands TB model ?

Response: We thank the reviewer for the excellent question. For clarity we answer in 2 parts. **Part 1** will detail a simple picture for how to understand the AHE in BTG (that of valley polarization) as discussed in the main text and **Part 2** will explain how this simple picture naturally captures the AHE phenomenology we observe, e.g., that of AHE arising at low carrier density and being turned on by displacement field.

Part 1: The anomalous Hall effect we observed in BTG (and its phenomenology, e.g., most pronounced at low density and turned on by displacement field) can be naturally understood to arise from a valley polarization in BTG. The essential ingredients of this are as follows:

- (i) A finite displacement field breaks inversion symmetry and allows the valleys (K and K') to possess Berry curvature, see e.g., Berry curvature distribution

Figure 3c in main text simulated from our 8 band model. The Berry curvature sign is opposite for opposite valleys (K vs K'). As a result, when time-reversal symmetry is preserved, the total net Berry flux (i.e. Berry flux in valley K + Berry flux in valley K') of the material vanishes and no anomalous Hall effect is detected.

- (ii) Close to regions of high density of states, valley symmetry can be spontaneously broken (e.g., through a Stoner-like ferromagnetism) where the distribution function in valley K and K' now become different: the state is valley polarized $f^K(\mathbf{p}) \neq f^{K'}(\mathbf{p})$. As a result, that Berry flux in valley K and valley K' are now imbalanced leading to a finite total net Berry flux and an anomalous Hall effect.

Part 2: Importantly, the distribution of the Berry curvature plays a significant role in the phenomenology of the AHE we observe. To see this, we first note that Berry curvature distribution in each valley is concentrated at the band edge (see Fig. 3c of main text). As a result, the sharpest change in Berry flux (as a function of chemical potential) occurs close to the band bottom too (see Fig. 3d of main text). As a result, even small valley polarizations ($f^K(\mathbf{p}) \neq f^{K'}(\mathbf{p})$) when chemical potential is fixed close to the band bottom (i.e. low density), can result in pronounced AHE. In contrast, when chemical potential is large (i.e. at high density) the change of Berry flux with chemical potential is small, making it harder to detect an AHE for modest valley polarizations. Second, we note that Berry curvature in each of the valleys depends on the application of a finite displacement field; Berry curvature vanishes when there is no displacement field. For very small displacement fields, the Berry curvature is highly concentrated at the band bottoms making access to the region with sharp change in Berry flux challenging. As a result, having a modest strength of displacement field is needed. This is consistent with our observation that AHE is turned on by the displacement field (see e.g., Supplementary Fig. 4 as well Supplementary Figure 8 of our revised manuscript) and is most pronounced for low densities.

To make the connection between the valley polarization and the observed AHE phenomenology in our devices stronger, we have included a discussion of how the observations of AHE being enhanced at low density and being turned on by D field is

directly connected and is consistent with the valley polarization and D-field induced Berry curvature distribution in BTG. This is now shown in the main text.

2) An half metallic state associated with a broken symmetry state breaking time reversal was also detected recently in few layers rhombohedral graphene (see for example <https://doi.org/10.1021/acs.nanolett.2c00466>). In that case it was attributed to a layer antiferromagnetic state. Could it be that some kind of magnetic state is present also here (with a weak magnetization) ? Can the author exclude in some way that the broken symmetry state is magnetic ?

Response: We thank the reviewer for the insightful question and the useful reference. We would like to answer the answer in two parts in which we will compare the main features of the magnetotransport data and magnetic state in Y. Lee et al Nano Lett. 22, 5094-5099 (2022) with that of our experiments on BTG, and comment on the state in the two mentioned cases.

Part 1: We first state the observations of the two experiments:

RFG: Y. Lee et al. Nano Lett. 22, 5094-5099 (2022) examines two-terminal magnetotransport measurements in rhombohedral few layer graphene (RFG) (consisting of both rhombohedral trilayers and rhombohedral tetralayers). The main features of the state they find are:

RFG 1. A large interaction induced gap of about 40 meV at charge neutrality in the absence of displacement field indicating a correlated insulating ground state in suspended RFG, see figure 1c in Y. Lee et al.

RFG 2. Hysteretic (two-terminal) magnetoconductance that are found only in rhombohedral stacked few layer graphene (see page 2 left column) when doped. Further, when undoped the insulating state is suppressed in the presence of a finite out of plane displacement field (see page 1, right column).

RFG 3. The hysteretic magnetoconductance is argued to be consistent with that of a spin-polarized half metal when the insulating layer antiferromagnetic (LAF) state is moderately doped (see figure 4b in Y. Lee et al). Note that spin-polarized half metals in rhombohedral trilayer graphene do not possess an AHE [as explicitly measured in Nature 598, 429-433 (2021)]

BTG: In our work, we study Bernal stacked tetralayer graphene (BTG) using a Hall-bar set-up. Spontaneously broken time-reversal symmetry is confirmed by a hysteretic AHE (with out-of-plane magnetic field) and is supported by quantum oscillations measurements. The main features of the state we find in our BTG devices are summarised again for comparison:

BTG 1: At zero displacement field we do not see any correlated gaps.

BTG 2: The state (characterized by an AHE) in our devices turns on at finite displacement field (see e.g., Supplementary Fig. 4). It is not suppressed at finite displacement field.

BTG 3: Since the state manifests as an AHE, its valley symmetry is broken (due to the opposite signs of Berry curvature in each valley). This valley polarized state is consistent with Stoner ferromagnetism.

Part 2: Comparison between magnetic state in RTG and BTG

Here we compare the results of the RFG in Y. Lee et al. Nano Lett. 22, 5094-5099 (2022) and our work on BTG. Here “C” refers to comparison.

C1: As seen above RFG1, RFG2, RFG3 directly contrast with BTG1, BTG2, BTG3 respectively. This contrast form the basis of the differences between the magnetic states in RFG and BTG which is further elaborated below. For instance, the large interaction induced gap in RFG1 of Y. Lee et al. Nano Lett. 22, 5094-5099 (2022) contrasts strikingly with the lack of any intrinsic gap (at zero displacement field) of our BTG devices.

C2: **Importantly**, the observation of AHE is a smoking-gun evidence of broken valley symmetry because opposite valleys have opposite signs of Berry curvature in the presence of a finite displacement field. Note because graphene has minimal spin-orbit coupling, opposite spin flavors have the **same** sign of Berry curvature. If the graphene multilayer structure is only spin polarized, an AHE is not expected. Indeed, a four-terminal measurement of a spin polarized half-metallic state in rhombohedral trilayer graphene was performed in Nature

598, 429-433 (2021) revealing a **vanishing** anomalous Hall effect. In summary, a finite AHE is enabled only if valleys are distinctly populated (valley polarization).

C3. If in addition to broken valley symmetry, spin is also polarized, quantum oscillations would show a *quarter* metal state. However, in the VM region, our quantum oscillations do not show evidence of a quarter metal state. Instead our quantum oscillations are consistent with a half metallic state (as discussed in the main text and shown in Figure 2).

C4. Additionally, we find that the AHE hysteresis in BTG persists to several tens of kelvin (see Figure 4 in main text). Whereas the hysteresis in two-terminal magnetoconductance disappear at a few Kelvin scale see (Y Lee et al., page 3 left column as well as abstract). This provides some indication that the energy scales and physics involved in the two devices are distinct.

C5: It is pointed out by Y. Lee et al. (see page 2, left column) that the magnetoconductance features they find are exclusive to rhombohedral stacks being minimal in Bernal stacks that they made.

On the basis of these comparisons (C1-C5) and on our experimental observations (e.g., BTG 1,2,3) we conclude that we have no evidence for spin polarization in our BTG devices. Further, the experimental phenomenology seen by Y. Lee et al. is quite different from what we see in BTG (we see no correlated gap, AHE is turned on at finite displacement field, hysteresis persists to several tens of kelvin). As a result, we also have no experimental evidence for an LAF state in our devices.

Instead, our measurements are most consistent with that of a spontaneously broken valley symmetry state and itinerant ferromagnetism. It can be understood (as has been the case with a number graphene multilayers and moiré systems, e.g., Ref. 6, 19, 22-24 of our main text) via a phenomenological Stoner ferromagnetism framework. To make the distinction between RFG and BTG clear, we have added a short description of the phenomenology of RFG spin polarized half metals in both Y. Lee et al. Nano Lett. 22, 5094-5099 (2022) [our Ref. 28] and Nature 598, 429-433 (2021) [our Ref. 6].

REVIEWERS' COMMENTS

Reviewer #1 (Remarks to the Author):

Authors have addressed all comments and answered questions properly in the revised manuscript and reply. I recommend its publication in Nature Communications.

Reviewer #2 (Remarks to the Author):

Reviewer #3 (Remarks to the Author):

The authors have addressed my comments and questions. I am excited to see that the AHE emerges for holes as well as electrons in this system. I recommend publication at this stage.

Reviewer #4 (Remarks to the Author):

Reviewer #5 (Remarks to the Author):

The authors correctly replied to all Referees' questions.

The paper can be published without any additional review.